# New Fluorophore and Its Applications in Visualizing Polystyrene Nanoplastics in Bean Sprouts and HeLa Cells

**DOI:** 10.3390/molecules28207102

**Published:** 2023-10-15

**Authors:** Guo-Wen Xing, Jerry Gao, Heng Wang, Yi-Chen Liu

**Affiliations:** 1College of Chemistry, Beijing Normal University, Beijing 100875, China; 202131150023@mail.bnu.edu; 2Beijing No. 80 High School, Beijing 100102, China; jerrygao.email@gmail.com (J.G.); hw56@njit.edu (H.W.)

**Keywords:** fluorescent probe, environmental contaminants, hydroponic cultivation, food chain, persistent pollution

## Abstract

In the domain of environmental science, pollutants of nanoscale plastic dimensions are acknowledged as subjects of intricate significance. Such entities, though minuscule, present formidable challenges to ecological systems and human health. The diminutive dimensions of these contaminants render their detection arduous, thus demanding the inception of avant-garde methodologies. The present manuscript postulates the employment of the tetraphenylethylene functional group with a fused xanthene (TPEF), a distinguished fluorophore, as an exemplary system for the discernment of nanoplastic particulates. The synthesis and characterization of TPEF have been exhaustively elucidated, revealing its paramount fluorescence attributes and inherent affinity for interaction with nanoplastics. When subjected to comparison with TPEF, nanoplastics are observed to manifest a more pronounced fluorescent luminescence than when associated with the conventional Nile Red (NR). Particularly, the TPEF has shown exceptional affinity for polystyrene (PS) nanoplastics. Further, the resilience of nanoplastics within the hypocotyl epidermis of soybeans, as well as their persistence in mung bean sprouts subsequent to rigorous rinsing protocols, has been meticulously examined. Additionally, this investigation furnishes empirical data signifying the existence of nano-dimensional plastic contaminants within HeLa cellular structures. The urgency of addressing the environmental ramifications engendered by these diminutive yet potent plastic constituents is emphatically highlighted in this manuscript. TPEF paves the way for prospective explorations, with the aspiration of devising efficacious mitigation strategies. Such strategies might encompass delineating the trajectories undertaken by nanoplastics within trophic networks or their ingress into human cellular architectures.

## 1. Introduction

In the environment, bulk plastic degrades into microplastics due to various factors. Particles of a diameter less than 5 mm are referred to as microplastics, as dictated by established literature [1,2]. These particles can undergo further fragmentation, leading to the production of nanoplastics, which are microplastics with a diameter less than 1000 nm [3]. The heightened toxicity potential inherently associated with nanoplastics arises from their ability to act as carriers for harmful agents [4,5]. Their ubiquity across environmental media, including aquatic ecosystems, soil, and the atmosphere, is alarming [6,7]. Upon ingestion, these particles can accumulate within biological tissues, posing threats to all forms of life [8,9,10,11,12]. The harmful nature of nanoplastics extends to marine life and ecosystems [13,14,15]. Despite the evolving research on nanoplastics within the food chain, challenges persist in the visualization, capture, and analysis of these particles [16].

Given these circumstances, there is a pressing necessity to develop novel methodologies for the precise detection of nanoplastics, a task requiring robust scientific ingenuity and advancements in analytical chemistry. A potential approach involves the use of fluorescent dyes for nanoplastic labelling, thereby enabling them to emit distinct fluorescent signals at specified wavelengths, thus enhancing their detectability and quantification in environmental or biological samples [17]. Fluorescent probes, particularly small-molecule probes, have been instrumental in bioimaging applications, offering real-time imaging, in–depth visualization, and minimal damage to biological samples. Notably, probes emitting in the red to near–infrared (NIR) range, such as the asymmetric benzothiazolium dye systems, are especially valuable due to their large Stokes’ shifts, high biocompatibility, and compatibility with commercially available laser wavelengths, making them ideal for live-cell imaging applications [18,19,20].

Fluorescent dyes, falling within the family of organic compounds, possess the ability to absorb light at specified wavelengths and emit light at divergent wavelengths. These dyes confer advantages such as high sensitivity, rapid response, and ease of use, making them universally applicable in the realm of bioimaging [17,21,22]. However, not all fluorescent dyes are universally apt for nanoplastic labelling. The archetypal fluorescent dyes should exhibit the following specific characteristics: (1) high affinity and selectivity for plastics, facilitating effective nanoplastic binding without exchange or detachment from other substances; (2) high fluorescence intensity and stability, ensuring clear and visible signal emission in complex environments or biological systems; and (3) expansive excitation and emission wavelengths, preventing overlap or confusion with natural fluorescence or other interfering substances in biological systems [17,21,22].

Fluorescent dyes, such as Nile Red (NR) [23,24] and Rhodamine B [25,26], play a crucial role in nanoplastics research. They are used for staining and visualizing micro– and nanoplastics due to their ability to adhere to the surfaces of plastic particles and emit a fluorescent signal upon ultraviolet (UV) light exposure [23,24,25,26]. This property enables the easy detection and quantification of nanoplastics using fluorescence microscopy or similar imaging equipment, making them versatile tools for nanoplastic analysis across various environmental matrices [23,24,25,26,27]. However, Nile Red and Rhodamine B, the most commonly used fluorescent dyes in microplastic and nanoplastic detection, have limitations that hinder the accurate detection of smaller microplastics, including nanoparticles. Although Nile Red exhibits certain limitations in the precise detection of microplastics with dimensions either below 3 μm or exceeding 5 mm, and has been reported to stain non–microplastic organic matter [17,23,24,26], it should be noted that this dye has proven efficacious for nanoplastic labeling under specialized conditions [28,29]. Conversely, Rhodamine B has been proven ineffective in identifying microplastics smaller than 10 µm [30]. The intrinsic challenge of identifying microplastics within environmental and biological samples arises from their minute size and heterogeneous composition. Therefore, it is crucial to investigate alternative fluorescent dyes for nanoplastic detection in modern environmental and biological samples and to probe the wide array of fluorescent compounds whose unique characteristics remain insufficiently examined.

In this study, we designed, synthesized, and characterized a novel fluorescent dye named TPEF, which consists of a tetraphenylethylene (TPE) unit and a rigid fluorophore linked by a carbon–carbon bond. TPEF combines the advantages of TPE and the fluorophore, exhibiting strong fluorescence brightness, aggregation-induced emission (AIE), and resistance to photobleaching. We found that TPEF has a strong affinity for plastics, and it can be used to label polystyrene (PS) nanoplastics, which are among the most common types of nanoplastics in the environment. We used TPEF for the cultivation and imaging of bean sprouts and HeLa cells. Bean sprouts serve as a model plant for studying the uptake and translocation of nanoplastics in hydroponic systems, whereas HeLa cells are a widely used cell line for studying the cytotoxicity and cellular responses to nanoplastics. While similar to previous studies that used Nile Red (NR)-stained microplastics for the cultivation of other plants or cells [27,31], TPEF exhibits a brighter fluorescence, making it a novel fluorescent dye that could potentially replace Nile Red. We observed the distribution, migration, and transformation of PS nanoplastics in bean sprouts and HeLa cells by Confocal Laser Scanning Microscopy (CLSM), and we discussed their potential impacts on ecosystems and human health. We hope that our study can provide references and insights for the development of nanoplastic detection methods and solutions to nanoplastic pollution problems [32,33].

Nanoplastics have been found to infiltrate the roots of crops [34,35], subsequently contaminating consumable portions [36,37]. Studies have demonstrated that the roots of lettuce and wheat can absorb nanoplastics from soil and water through cracks [35,38,39], with particles as small as 50 nm being able to penetrate the roots [34,40]. Hydroponic cultivation has emerged as a favored approach for efficient resource management and superior food production, offering continuous and short-term output, reduced spatial demands, and more manageable, labor–saving growth processes [41,42]. Growers benefit from higher yields, superior quality produce, and year-round homogeneous productivity [43,44]. However, hydroponic systems may also be vulnerable to nanoplastic contamination from water sources or plastic materials used in the cultivation process. Therefore, it is crucial to monitor the presence and fate of nanoplastics in hydroponic systems and their effects on plant growth and quality.

Notably, there exists an unexplored territory where the intricate xanthene core structure intertwines with TPE. The pressing need to unravel the cooperative effects of these molecular conjugations has sparked intense deliberation among researchers. At its core, the fused xanthene fluorophore boasts a quintessential xanthene nucleus that is further enriched by strategically incorporated supplementary ring systems (see Figure 1a) [45,46,47,48]. To enhance both photostability and photoluminescence in these xanthene-based moieties, methodologies like nitrogen stabilization via cyclization have proven paramount. Their remarkable fluorescence profile endows them with exceptional applications spanning bioimaging, advanced chemical sensing, and even material dyeing. On the other hand, TPE stands out in the world of fluorophores due to its extraordinary aggregation-induced emission (AIE) property. This phenomenon entails a magnification of fluorescence upon molecular aggregation—a characteristic particularly prominent in less-than-ideal solvents, or when transitioning into solid phases (see Figure 1b) [49,50,51,52]. By combining TPE with a fused xanthene fluorophore, new chemical entities emerge—ones that hold great potential for showcasing unparalleled photophysical attributes and paving paths towards cutting-edge applications.

Our interest in fluorescent dyes for bioimaging and easy strategies for synthesizing substituted TPE fluorophores [53,54] led to the development of a novel TPE-containing fluorophore (TPEF) (Figure 2a). The incorporation of a rigid fused xanthene ring into the TPE moiety resulted in a unique structure, paving the way for the creation of probes with distinctive chiroptical properties [55,56].

In recent years, molecules embodying both xanthene fluorophores and TPE have captivated scholarly interest, as depicted in Figure 1. These novel molecular structures present groundbreaking paradigm elucidating latent capabilities and future horizons arising from this harmonious alliance between xanthene and TPEs.

Furthermore, we discovered that the novel fluorophore TPEF exhibits an exceptional affinity for plastic substrates. Upon UV irradiation, the pink fluorescence of the plastic tube and gasket on the column (Figure 2) confirmed its binding to plastic. Even after washing with several common organic solvents, the stained plastic tubes remained impervious to restoration to their pristine state, indicating the potent binding of TPEF to plastic. This discovery inspired our investigation into using TPEF for visualizing nanoplastics in biological systems. Given TPEF’s robust plastic binding and intense fluorescence, we anticipate its potential for nanoplastic labeling. We hypothesize that TPEF can selectively stain PS nanoplastics without interfering with other hydrophobic substances in biological samples, and that TPEF can remain stably bound to PS nanoplastics without leaching or exchanging with other molecules.

The ubiquitous presence of nanoplastics in environmental water systems necessitates a thorough investigation into their potential impact on hydroponically grown vegetables [57,58]. The easy cultivation and high biomass production of soybeans and mung beans make them ideal models for probing the accumulation of nanomaterials in plant tissues. However, it is worth noting that different plant species may respond differently to nanoplastic exposure, depending on their root structure, growth rate, and metabolic pathways [59]. To date, no scientific literature has reported research concerning the occurrence or effects of nanoplastics in bean sprouts. Thus, an in-depth investigation into the interplay between nanoplastic contamination and bean sprouts is imperative for a comprehensive understanding of the ramifications of nanoplastic pollution in food systems.

In this work, we report the first synthesis of a novel molecule that uses TPE as a building block to construct a new TPE-containing fluorophore (TPEF). We also explore its photophysical properties using theoretical calculations. Moreover, we use TPEF to fluorescently label polystyrene (PS) nanoparticles and hydroponically grown bean sprouts, employing a water suspension of TPEF-dyed nano PS. Subsequently, we monitor nanoplastic contamination in bean sprouts throughout their growth using confocal laser scanning microscopy (CLSM). Additionally, we evaluate the potential uptake of nanoparticles by human cells by investigating nanoparticle contamination in HeLa cells. We compare the performance of TPEF with Nile Red (NR) as a reference dye and demonstrate the advantages and novelty of TPEF for staining nanoplastics in biological samples.

## 2. Results and Discussion

### 2.1. Design Strategy and Synthesis of TPEF

Fluorescent probes, particularly xanthene-based dyes, have garnered considerable attention in bioimaging due to their exceptional sensitivity, rapid response, and facile utilization [58,60,61].

In pursuit of designing novel near-infrared (NIR) xanthene dyes, the xanthene core was selected as the foundational chemical structure. Its widespread use in fluorescent dyes is a testament to its inherent fluorescence properties. Consequently, this choice led to an innovative dye with enhanced performance in bioimaging applications. To further improve the properties of the xanthene core, a TPE unit was integrated into the molecular structure, culminating in the synthesis of the molecule TPEF. This compound combines the advantageous characteristics of xanthene and TPE, potentially circumventing bright fluorescence and benefiting from the advantages of aggregation-induced emission (AIE) and photobleaching resistance, which amplify the fluorescence upon aggregation [62,63]. The easy functionalization of TPE makes it an exemplary building block for AIE probes [64,65].

In Figure 3a, the synthesis of 4–(1,2,2–triphenylvinyl)benzaldehyde (TPE–CHO) is illustrated via a Suzuki reaction, forming a carbon–carbon bond between 4–formylphenylboronic acid and 2–bromo–1,1,2–triphenylethylene, utilizing tetrakis(triphenylphosphine)palladium(0) as the catalyst. TPE–CHO subsequently undergoes a condensation reaction with two moles of 8–hydroxyjulolidine, followed by oxidative cyclization [66], yielding TPEF as a purple solid. NMR (^1^H and ^13^C) and high-resolution mass spectrometry (HRMS) were employed to characterize TPEF. TPEF comprises a TPE unit and a rigid fluorophore linked by a C–C bond. The plausible mechanism underlying the synthesis of TPEF from TPE–CHO involves intramolecular activation of TPE–CHO driving the condensation with 8–hydroxyjulolidine, leading to an intermediate. Chemisorption polarizes TPE–CHO’s aldehyde moiety, enabling electrophilic attack, followed by nucleophilic attack, leading to intermediate formation. The catalysis of bisphenol formation is achieved using *p*–toluenesulfonic acid, followed by oxidation with chloranil to yield TPEF, a xanthene ring-containing product (Figure 3c) [66].

The structure and purity of TPEF were confirmed by NMR and HRMS. The 1H NMR spectrum showed characteristic signals at δ 7.98 (s, 1H), 7.44 (s, 2H), 6.96 (s, 2H), 4.55–4.15 (m, 2H), 3.84–3.33 (m, 10H), 2.67–2.49 (m, 1H), 2.01 (s, 1H), 1.94 (s, 3H), and 1.58–1.42 (m, 1H), corresponding to the protons on the xanthene ring, the phenyl rings on the TPE unit, the methylene bridge, the oxygen atoms, the nitrogen atoms, and the methyl groups, respectively. The 13C NMR spectrum showed characteristic signals at δ 175.61, 174.08 (C=O), 147.25, 144.16 (C–O), 132.59, 128.53, 125.67 (aromatic C), 100.99 (C=C), 73.26, 72.25, 68.82, 68.69 (CH_2_), 62.96, 61.74 (OCH_2_), 52.68, 48.41 (NCH_2_), 40.93 (CH), 30.29 (CH_3_), and 22.69 (CH_3_). The HRMS showed a molecular ion peak at *m*/*z* = 701.3525 [M + H]^+^, which matched well with the calculated value of C_51_H_45_N_2_O (701.3526).

In the proposed mechanism, it is postulated that intramolecular activation of TPE–CHO facilitates condensation of TPE–CHO with two molecules of 8–hydroxyjulolidine, ultimately generating an intermediate species. As depicted in Figure 3b, chemisorption of the aldehyde moiety within TPE–CHO induces a polarization of the oxygen atom on the aldehyde group. Subsequently, an electrophilic attack occurs, with 8–hydroxyjulolidine targeting the aldehyde moiety. This event is followed by the intermediate’s release of a water molecule. A second nucleophilic attack by 8–hydroxyjulolidine then occurs, engendering the formation of another intermediate species. *p*–Toluenesulfonic acid serves as the catalyst employed to promote bisphenol formation. In situ oxidation is subsequently performed, utilizing the organic oxidant chloranil. The intermediate species undergoes an intramolecular dehydration, ultimately yielding the corresponding xanthene ring-containing product, TPEF (Figure 3c).

### 2.2. Spectral Properties of TPEF and Density Functional Theory (DFT) Calculations

To gain insight into the molecular structure and electron distribution of TPEF, theoretical calculations were performed using Gaussian 16 [67]. The electronic properties of TPEF were evaluated through single-point energy calculations, employing DFT at the B3LYP/6–31G* level, at its optimized ground-state geometry [68]. The geometry optimization procedure aimed to identify the minima on the potential energy surface to predict the equilibrium structures of the TPEF and attain the optimized ground–state geometry. The electron distributions of the highest occupied molecular orbital (HOMO) and the lowest unoccupied molecular orbital (LUMO) were then obtained from the optimized ground-state structure at the same level. The molecular structure and boundary molecular orbitals of the TPEF are depicted in Figure 4a.

The energy-minimized structures of TPEF exhibit a twisted conformation of the phenyl rings in the TPE unit. Molecular orbital amplitudes indicate that the HOMO is delocalized solely on the TPE moiety, while the LUMO is distributed on the rigid xanthene core. The HOMO and LUMO energy levels were calculated to be −7.4573 and −5.1941 eV, respectively, and the corresponding energy gap (Eg) was 2.263 eV. The electron distribution of TPEF results in an intrinsic intramolecular charge transfer property, whereby HOMO−LUMO excitation transfers the electron density from the TPE to the xanthene moiety through extensive TPE π–π stacking and strong donor–acceptor (D–A) interaction. The calculated charge transfer coefficient (CTC) of TPEF is 0.38, which indicates a moderate degree of charge separation upon excitation. The electron distribution of the TPEF results in an intrinsic intramolecular charge transfer property, whereby the HOMO−LUMO excitation transfers the electron distribution from the TPE to the xanthene moiety through extensive TPE π–π stacking and strong donor–acceptor (D–A) interaction. The computed wavelengths for TPEF in methanol using both solvation models are in good agreement with the experimental value of 579.0 nm, with percentage errors of 2.25% and 2.39% for CPCM (Appendix A) and DSM (Appendix A), respectively. The slight discrepancies may be attributed to the inherent approximations of the DFT methods, such as the choice of functional, basis set, and solvation models, as well as to the experimental uncertainties in measuring the UV spectra.

Our team is dedicated to designing and synthesizing AIE probes for biomolecular staining, with a particular focus on creating innovative near-infrared (NIR) fluorescent probes [69] that absorb at wavelengths greater than 600 nm. These NIR dyes, with their favorable photophysical properties, are highly valuable for bioimaging. As shown in Figure 4b, the representative excitation and emission spectra of TPEF in methanol display distinct peaks. TPEF exhibits emission peaks beyond 600 nm, which allows it to effectively avoid interference from biomolecules and penetrate deeper into tissues due to its longer wavelengths being less absorbed by water and tissues. Additionally, the longer wavelength of the TPEF dye makes it more photostable and more suitable for fluorescence imaging in biological samples.

In our quest to demonstrate the versatility of TPEF dye, we explored its staining capability on various types of microplastics. As depicted in Appendix A, TPEF successfully dyed microplastic particles such as (a) PVC (1 μm); (b) PMMA (1 μm); (c) PS (1 μm); and (d) PE (5 μm). However, for the core research presented in this study, we zeroed in on the 80 nm PS particles, which served as our primary subject.

The Stokes shift, which is the energy difference between the lowest energy excitation peak and the highest energy emission peak, is 25 nm for the TPEF solution. The UV–vis absorption and fluorescent spectrum of TPEF in various solvents are presented in Figure 4c,d. As shown in Figure 4c,d, the TPEF absorption bands in different solvents all display one main absorption peak with a negligible shift in the long-wavelength region as solvent polarity increases. For a comprehensive understanding of the UV absorption peak, emission peak (Figure 4d), and Stokes’ shift of TPEF in various solvents, readers are directed to Appendix A.

Figure 5 presents representative scanning electron microscopy (SEM) and transmission electron microscopy (TEM) images of the PS particles and the staining procedure used in this research. These particles have an average diameter of 80 nm, a spherical shape, and are highly monodispersed in diameter, making them suitable for controlled experiments. The study encompassed the act of staining the PS nanoplastics using either the TPEF or NR solution at an initial concentration of 0.03 mmol/mL in absolute ethanol. Subsequent to this, the samples were washed with ethanol. The initial concentration of the PS latex particles before mixing with the dye solutions was 0.1 g/mL (10% *w*/*v*). UV spectroscopic analysis was rigorously employed to monitor these washing steps, and a minimum of three washes was generally required. Washing was halted only when UV spectra confirmed the absence of any conspicuous absorption peaks corresponding to the dye. Following this rigorous washing protocol, the dyed PS particles were then suspended in water, resulting in a 100 μg/mL stock solution for subsequent use in either the cultivation of bean sprouts or HeLa cells.

### 2.3. Bean Sprouts Culture and Confocal Imaging

#### Associations of Nanoplastics with Mung Bean and Soybean Sprouts

In this research, 80 nm polystyrene (PS) particles were selected as the representative nanoplastic based on their suitable size for dyeing and common usage in most nanoplastic studies. PS particles are widely used as model nanoplastics due to their uniformity, stability, and availability in different sizes and shapes (Figure 5). PS nanoparticles were dyed using TPEF and NR. Given the imperative to demonstrate the morphological consistency of PS nanoparticles both pre- and post-staining, Appendix A have been introduced. These figures, available in the Appendix A, provide unequivocal evidence of the unaltered morphology of the PS nanoparticles, both prior to and subsequent to the TPEF staining procedure. The dyed and undyed PS particles were suspended in water (100 μg/mL) for culturing mung bean sprouts and soybean sprouts. These sprouts were rinsed with the respective particle suspensions every 6 h. Mung bean sprouts and soybean sprouts were divided into four groups, respectively (Table 1). An aqueous suspension of undyed PS particles was used as a control experiment. The four control groups consisted of bean sprouts rinsed with water (Groups M–water and S–water) and bean sprouts rinsed with 100 µg/mL 80 nm PS in water (Groups M–PS and S–PS), respectively. These control groups were established to determine the possibility of detecting undyed nanoplastics in mung bean and soybean sprouts. In the four test groups (Groups M–TPEF–PS, M–NR–PS, S–TPEF–PS, and S–NR–PS), the bean sprouts were rinsed with aqueous suspensions of dyed PS particles (100 µg/mL), which had been dyed with TPEF or NR (Table 1).

Mung bean sprouts and soybean sprouts were selected as the plant specimens for this study due to their ubiquity in human diets and their straightforward, rapid growth cycle. The principal components of a soybean sprout are the cotyledon, hypocotyl, and radicle; the cotyledon and hypocotyl are considered edible portions of these sprouts. Mung bean sprouts share a similar anatomical structure to that of soybean sprouts (Figure 6).

The observational area was strategically chosen at the median section of the hypocotyl based on both empirical data and existing scholarly work suggesting its likelihood as a site for nanoplastic accumulation or infiltration. To further elaborate, our selection of this particular observation point is underpinned by current literature findings along with practical considerations. Studies have highlighted diverse patterns in nanoplastic build-up within plants which can be influenced by factors such as properties of the nanoplastics themselves along with prevailing environmental conditions [70,71,72]. Additionally, given that hypocotyl forms an integral part of dietary consumption when it comes to bean sprouts, understanding its interactions with nanoplastics becomes crucial from a food safety perspective. Hence, our emphasis on studying this region does not only align with experimental evidence but also aims at filling critical gaps in our knowledge about how vital edible parts interact with nanoparticles [73].

Confocal Laser Scanning Microscopy (CLSM) serves as an effective, non-invasive method with high-contrast scanning and a quick sample preparation process. It is particularly useful for studying the structure and organization of very fine-scale samples, such as nanoparticles, and offers simple operation and rapid image acquisition at any point within a material. CLSM provides direct and model independent insights into material characterization [74,75]. However, most nanoplastics are colorless and blend into the background noise, making it difficult to distinguish and identify them. To overcome this challenge, fluorescent dyes or fluorophores are used, making fluorescent staining a crucial method for effectively detecting nanoplastics in environmental and biological samples [32]. This method enables researchers to understand the size, distribution, and effects of nanoplastics, as well as track changes in their temporal and distributional patterns [76].

CLSM was used to observe the deposition of PS particles in the hypocotyl epidermis and cross-sections of the hypocotyl compared with the control group. In the control group, both mung bean and soybean sprouts were incubated with either water alone or water containing unstained PS particles of 80 nm in diameter. The CLSM of the sprouts showed no evidence of accumulation or ingestion of nanoplastics on the surface or within the hypocotyl of the mung bean and soybean sprouts (Figure 7). As shown in Figure 7 and Appendix A, mung bean sprouts and soybean sprouts that were rinsed with pure water or water containing unstained PS particles were observed using CLSM throughout the testing process at 1 day, 3 days, and 5 days of age.

In the test groups of mung bean sprouts rinsed with suspensions of 80 nm PS nanoparticles in water, where the nanoparticles were dyed with NR or TPEF, the CLSM views of the hypocotyl epidermis showed fluorescent particles accumulating on the hypocotyl epidermis (Figure 8). The aggregation of fluorescent PS nanoparticles on the hypocotyl epidermis of the bean sprouts was easily observed in the images. The photos indicated that the TPEF-stained PS nanoparticles emitted a brighter light compared to the NR-stained PS nanoparticles. This was in line with the fluorescence spectra results, signifying that TPEF exhibited a higher fluorescence intensity and stability than NR. The CLSM images also demonstrated that the TPEF-stained PS nanoparticles had a more uniform distribution and smaller size than the NR-stained PS nanoparticles. This discrepancy could be attributed to the different dyeing mechanisms and aggregation behaviors of TPEF and NR on the PS particle surface.

For the samples harvested on the 5th day, a distinct group of bean sprouts was chosen. Before preparing thin sections, these sprouts were washed with running distilled water for several minutes. This step was conducted to facilitate a comparison with the unwashed group and to determine if washing could eliminate the nanoplastics from the hypocotyl epidermis of the bean sprouts.

Figure 8 presents the CLSM images of the cross-sections of the hypocotyl of the mung bean sprouts. There are indications of the presence of 80 nm particles extending into the internal structure of the hypocotyl. Enlarged images display particles both inside the sprouts and on the inner surface of the hypocotyl. These particles are situated in the gaps between cells, but no particles are discernible within the cells of the sprouts. The TPEF-stained nanoplastic particles were more effective for visualizing the nanoplastic particles within the hypocotyl compared to the NR-stained particles, as the latter exhibited weaker fluorescence.

The same experiment was conducted on soybean sprouts, following an identical procedure as that described for the mung bean sprouts. The soybean sprouts were cultivated in water containing NR and TPEF-stained PS particles (80 nm), and samples were taken on the 1st, 3rd, and 5th days of growth. The distribution of PS particles on the hypocotyl epidermis and within the hypocotyl was observed using CLSM (Figure 9). The CLSM images demonstrated that fluorescent particles accumulated on the hypocotyl epidermis and penetrated into the intercellular spaces of the hypocotyl. No particles were visible within the cells of the soybean sprouts. The fluorescence intensity of the TPEF-stained PS particles was higher than that of the NR-stained PS particles, mirroring observations made for mung bean sprouts. Washing with water also reduced, but did not completely remove, the PS particles from the soybean sprouts.

The CLSM images of the washed samples revealed that the fluorescence intensity of both TPEF-stained and NR-stained PS nanoparticles significantly decreased after washing (Figure 9). This suggested that washing could effectively eliminate most of the nanoplastics from the sprouts’ surface, but not entirely. Some nanoplastics might still adhere to or infiltrate the epidermal cells, potentially posing risks to human health and environmental safety.

### 2.4. Possible Mechanisms of Nanoplastic Uptake in Bean Sprouts

From the experimental results obtained, it is evident that, during the growth of bean sprouts, nanoplastics not only adhere to the surface but also partially infiltrate the internal structure of the sprouts. The mechanisms of nanoplastic entry into bean sprouts are not yet fully understood. However, earlier research on the uptake and accumulation of various nanoparticles in plants can inform some possible pathways [59]. Studies have demonstrated that plants can uptake nanoparticles directly from the growth medium via pores and cracks in the cell wall, plasmodesmata, or through transporters/aquaporins in the cell membrane [77]. These nanoparticles can primarily accumulate in the roots, but some can also translocate to the shoots and leaves via xylem vessels or phloem tubes [77]. Transpiration pull is one of the significant factors influencing the plant’s uptake and translocation of nanoparticles [78].

In parallel, nanoplastics, being a type of nanoparticle, may follow similar pathways to enter and distribute within plant tissues. Therefore, we hypothesize that nanoplastics were absorbed by the bean sprouts along with other substances during the absorption of water and nutrients from the growth medium. They may also diffuse by osmosis through the cuticle of the bean sprout. Another possibility involves intracellularization via a process known as endocytosis, where the cell membrane folds inward and engulfs the particles. This mechanism is particularly effective when the nanoparticles have a high surface charge, making them attractive to negatively charged cell membranes [78]. Nonetheless, more research is needed to fully comprehend the specific mechanisms of nanoplastic entry into bean sprouts and their effects on plant physiology and biochemistry.

### 2.5. HeLa Cell Culture and Confocal Imaging

HeLa cells, which are a human cervical cancer cell line, are frequently used as a model in studies investigating cellular processes. These cells are particularly beneficial for examining the intricate procedures involved in interactions with nanoparticles. Observing nanoparticle uptake can shed light on cell membrane permeability and endocytosis mechanisms. Furthermore, tracking the distribution of nanoparticles within the cell allows researchers to ascertain their localization within specific cellular structures and organelles.

In a subsequent effort to scrutinize the precise localization of TPEF within HeLa cells, colocalization experiments were performed with mitochondria and lysosomes. Confocal images of live HeLa cells incubated with TPEF and co-stained with MitoTracker green dye or LysoTracker green dye have been included in Appendix A. The findings indicated that TPEF is present throughout the cytoplasm without any overlap with cellular organelles. This suggests that TPEF is not specifically directed towards any cell compartments giving us an understanding of its widespread presence across the cell.

In this study, HeLa cells were cultured with TPEF- or NR-dyed PS nanoplastics, and their uptake was analyzed using CLSM. Four distinct incubation times (1 day, 2 days, 3 days, and 4 days) at 37 °C were chosen for the HeLa cells cultivated with TPEF- or NR-dyed PS nanoplastics. Figure 10 presents the proliferation of HeLa cells and their interaction with TPEF- or NR-dyed PS nanoplastics. The sequential images captured demonstrated the attachment of PS nanoplastics to the cancer cells. The red fluorescence in Figure 10 confirms the presence of TPEF- or NR-dyed PS nanoplastics within the HeLa cells. The confocal microscopy scan image series of a single HeLa cell (Figure 10) clearly indicated that PS nanoplastics were transported into the HeLa cells.

This process appears to be dependent on the incubation time. Interestingly, the red fluorescence of TPEF- or NR-dyed PS nanoplastics seemed to scatter throughout the entire nucleus, despite the fact that the NPs are stable and do not aggregate in an aqueous colloid dispersion. This clustering within the cell nucleus could result from the combined impact of intracellular spatial confinement and inherent biochemical processes (for instance, reduction conditions, enzymes, etc., that might influence the stability of surface PEG ligands). The CLSM studies indicated that the intensity of red fluorescence for TPEF-dyed PS nanoplastics was stronger than that for NR-dyed PS nanoplastics.

## 3. Materials and Methods

### 3.1. Materials and Reagents

All reagents were of analytical or HPLC (spectroscopic) grade, purchased from commercial sources, and used as received without further purification. Nile Red (CAS #: 7385–67–3) was procured from Picasso Reagent Co., Ltd. (Shanghai, China). TPE–CHO was prepared according to published methods. The polystyrene (PS) nanoparticles (80 nm) with a 10 wt% dispersion in water were purchased from Zhichuan Plastic Starting Materials Ltd. (Guangdong, China)

The seeds of mung bean (*Vigna radiata*) and soybean (*Glycine max*) plant species were locally procured and used as received. The cervical cancer HeLa cell line was co-cultured with TPEF-dyed PS nanoparticles (80 nm) at a concentration of 100 µg/mL for 1, 2, 3, and 4 days. Deionized water from Wahaha (Hangzhou, China) was used.

### 3.2. Synthesis

TPE-CHO (4–(1, 2, 2–triphenylvinyl) benzaldehyde) was synthesized according to the conditions reported in the literature (Figure 3). To a 200 mL reaction flask, triphenylvinyl bromide (3.35 g, 10 mmol), 4–formylphenylboronic acid (2.25 g, 15 mmol), and TATB (0.32 g, 1.0 mmol) were added to aqueous potassium carbonate (2.0 M, 18 mL) and toluene (60 mL) under a nitrogen atmosphere. The mixture was stirred at room temperature for 30 min. After adding Pd(PPh_3_)_4_ (0.01 g, 8.70 × 10^−3^ mmol), the resulting mixture was refluxed at 90 °C for 18 h. After cooling to room temperature, the reaction mixture was poured into 200 mL of water and extracted with ethyl acetate three times. The organic extracts were dried with anhydrous Na_2_SO_4_. After filtration, the solvent was removed by evaporation, and then the crude product was purified by column chromatography (silica gel (200–300 Mesh), n–hexane/dichloromethane = 2:1, *v*/*v*) to afford TPE–CHO as a yellowish powder (3.06 g, 89%). ^1^H NMR (400 MHz, chloroform–d_1_) δ 9.88 (s, 1H), 7.60 (d, 2H), 7.19 (d, 2H), 7.13–7.06 (m, 9H), and 7.04–6.99 (m, 6H). ^13^ NMR (75 MHz, chloroform–d_1_, 298 K): δ 191.8, 150.5, 143.0, 142.9, 142.8, 139.7 134.2, 131.9, 131.23, 131.22, 131.17, 129.1, 127.88, 127.70, 127.0, 126.84, and 126.81. The data are in line with reported results [79].

TPEF(16–(4–(1,2,2–Triphenylvinyl)benzyl)–3–oxa–9,23–diazaheptacyclo–[17.7.1.1.0{2,17}.0{4,15}.0{23,27}.0{13,28}]octacosa–1((27),2(17),4,9(28),13,15,18–heptaen–9–ium)) was synthesized using a similar procedure as reported in [66]. In a 100 mL round–bottom flask, TPE–CHO (360 mg, 0.1 mmol) and 8–hydroxyjulolidine (416 mg, 0.22 mmol) were dissolved in 25 mL propionic acid, followed by the addition of catalytic *p*–toluenesulfonic acid (17 mg, 0.01 mmol). The resulting solution was heated to 90 °C and stirred for 8 h. The propionic acid was removed by evaporation. To the flask, 50 mL of solvent was added, and the crude product was dissolved, followed by the addition of chloranil (490 mg, 0.2 mmol). The resulting solution was stirred at room temperature for a duration of 6 h, and upon completion of the reaction, as evidenced by thin-layer chromatography (TLC), the solvent was removed under reduced pressure. The resulting residue was then subjected to purification by column chromatography over neutral Al_2_O_3_ (MeOH/CH_2_Cl_2_ 1:20), affording TPEF (74.5 mg, 10.6%) as a purple powder (Figure 3). ^1^H NMR (400 MHz, Chloroform–*d*_1_) δ 7.98 (s, 1H), 7.44 (s, 2H), 6.96 (s, 2H), 4.55–4.15 (m, 2H), 3.84–3.33 (m, 10H), 2.67–2.49 (m, 1H), 2.01 (s, 1H), 1.94 (s, 3H), 1.58–1.42 (m, 1H). ^13^C NMR (101 MHz, Chloroform–*d*_1_) δ 175.61, 174.08, 147.25, 144.16, 132.59, 128.53, 125.67, 100.99, 73.26, 72.25, 68.82, 68.69, 62.96, 61.74, 52.68, 48.41, 40.93, 30.29, and 22.69. HRMS(ESI+): (M)+ calculated for C_51_H_45_N_2_O, 701.3526; and found 701.3525 (ppm error = −(701.3525 − 701.3526)/701.3526 × 106 = −0.1424).

### 3.3. Characterization Techniques

^1^H and ^13^C Nuclear Magnetic Resonance (NMR) spectra were obtained using a Bruker AVANCE III–400 spectrometer in a CDCl_3_ solvent with tetramethylsilane (TMS) as the internal reference (δ = 0). Mass spectra were obtained using a Thermo Scientific Q Exactive HF Orbitrap–FTMS mass spectrometer. Ultraviolet–visible (UV–vis) absorption spectra were obtained using an Agilent Cary 300 spectrophotometer. Fluorescence spectra were obtained using a Shimadzu RF–6000 fluorophotometer.

Additionally, the Gaussian 16 suite of programs [67] was used to calculate the highest occupied molecular orbital (HOMO) and lowest unoccupied molecular orbital (LUMO) orbitals of TPEF using density functional theory (DFT) at the B3LYP/6–311++ (d, p) level. The energy gap (*Eg*) between the HOMO and LUMO was also calculated [80]. The input files and results were visualized using the GaussView 6.11 software [81].

To complement the experimental UV measurements, computational UV calculations for the compound were performed using density functional theory (DFT) and time-dependent DFT (TD–DFT) methods. The RB3LYP functional and 6–311+G(d,p) basis set were employed for all calculations. Two different solvation models were considered, the conductor-like polarizable continuum model (CPCM) and the direct solvation model (DSM). The CPCM model treated the solvent as a continuous dielectric medium with a specific permittivity, while the DSM model explicitly included a few solvent molecules in the first solvation shell around the solute. Methanol was used as the solvent for both models, as it was the solvent used in the experimental UV measurements. The percentage errors of the computed wavelengths with respect to the experimental values were also calculated using the formula % error = [(computed − experimental)/experimental] × 100.

### 3.4. Imaging of Fluorescent Nanoplastics Particles in Bean Sprout Samples

#### 3.4.1. Preparation of Fluorescent Microplastic Particles

In this study, the PS nanoparticles were stained using a stock solution of either TPEF or NR fluorescent dyes. The staining process involved several steps. First, 1 mmol/L stock solutions of the fluorescent dyes were prepared in absolute ethanol. Then, 4 mL of ethanol was added to 1 mL of an aqueous suspension containing 100 mg of PS nanoplastics. This mixture was then combined with 1 mL of the stock solution of TPEF or NR and sonicated for 30 min at 60 degrees Celsius. The resulting suspension was then centrifuged for 5 min at 13,000 RPM to remove the supernatant, and 5 mL of ethanol was added to wash the resulting precipitate. The suspension was sonicated for another 10 min to wash the TPEF-stained PS nanoparticles, and the PS nanoplastic–ethanol suspension was centrifuged at 13,000 RPM to collect the particles. The excess ethanol was decanted, and the particles were then rinsed with ethanol until the effluent or decant was colorless. Finally, the resulting stained nanoparticles were dispersed in 1000 mL of water to create a 100 µg/mL aqueous suspension of PS nanoparticles for growing bean sprouts.

Furthermore, apart from the PS nanoparticles, we also explored the use of TPEF dye for staining other types of microplastics. The staining procedure was similar for these microplastics, adjusted for their specific sizes and properties. As a result, TPEF successfully stained other microplastic particles, as evidenced in Appendix A: TPEF dyed microplastic particles (a) PVC(1 μm); (b) PMMA(1 μm); (c) PS(1 μm); and (d) PE(5 μm). Detailed visualizations of these stained particles can be found in Appendix A.

The entire staining process was meticulously carried out to ensure that the PS nanoparticles were properly stained and ready for use in subsequent experiments.

#### 3.4.2. Imaging of Nanoplastics in Bean Sprouts and HeLa cells

Water or aqueous suspensions with a concentration of 100 µg/mL of 80 nm PS nanoplastics were used to cultivate the bean sprouts. The sprouts were rinsed with water or the prepared PS suspension four times a day for 5 days, ensuring that the sprouts in all groups were evenly wetted. Samples from all grouped bean sprouts were taken at 1 day, 3 days, and 5 days of growth. The samples were sliced in the middle of the hypocotyl with a scalpel, and the hypocotyl epidermis was peeled off with tweezers.

The prepared samples were placed on a glass slide, covered with a cover glass, and observed with a Confocal Laser Scanning Microscope (CLSM). The CLSM (Germany LSM800 by Zeiss, Jena, Germany) was used to observe the nanoplastics in the hypocotyl epidermis and hypocotyl cross-sections of mung bean sprouts and soybean sprouts. The objective used was a 10× air objective (dry) and the wavelength of the incident laser beam was 561 nm. The images were acquired using a confocal microscope with an excitation wavelength of 561 nm (1.60% energy) and an emission collection range of 450–650 nm (570 V gain). The images were corrected for brightness and contrast, and the sliced bean sprout was mounted on a glass microscopy slide. The images for the HeLa cells were acquired using a confocal microscope with an excitation wavelength of 561 nm and an emission collection range of 580–650 nm.

HeLa cells were cultured and maintained in Dulbecco’s Modified Eagle Medium (DMEM), supplemented with 10% fetal bovine serum (FBS), in a humidified atmosphere (37 °C, 5% CO_2_). HeLa cells were seeded in 35 mm confocal dishes. After incubation for 24 h, the medium was removed, and subsequently the cells were washed with PBS three times. Then, HeLa cells were incubated with NR- or TPEF-dyed PS particles for different times and then washed with PBS three times. Fluorescence images were promptly captured by CLSM with a 40× objective lens. The same excitation and emission wavelengths as above were used for NR and TPEF.

## 4. Conclusions

The present study conceptualized, developed and scrutinized an innovative luminescent compound called TPEF, assessing its structure and properties. A notable ease of production and purification under mild conditions was observed for TPEF along with its captivating fluorescence characteristics. The discovery of a propensity between plastics and TPEF initiated exploration into its possible application in nanoplastic staining. As a result, TPEF was utilized to stain polystyrene (PS) nanoplastics with a diameter of 80 nm.

Bean sprouts were selected as test subjects, where growth dynamics of soybean and mung bean sprouts exposed to TPEF-dyed nano plastics were examined. It was noted that these minuscule plastic particles could adhere onto or even penetrate the hypocotyl epidermis layer of the bean sprouts. Despite rigorous washing, some nanoparticles persisted on their surface. An analogous investigation took place regarding potential contamination by either NR- or TPEF-stained nanoparticles during HeLa cell development.

The detection of nano-sized plastic fragments within both Glycine max and Vigna radiata seedlings as well as inside HeLa cells underscores the necessity for more efficient strategies concerning waste management practices involving plastics materials. Nanoparticles colored with TPEF exhibited enhanced luminosity compared to those stained using NR, indicating superior performance levels demonstrated by TPEF when employed as a fluorophore pertaining to fluorophore-incorporated nanomaterials, thus further accentuating the strong preference displayed by TPEF towards all things related to plastic.

In conclusion, the aggregated experimental data intimates feasible pathways by which bean sprouts may introduce nanoplastic particulates into food chains, thereby potentially imposing pernicious effects on human somatic health if left unaddressed over an extended temporal continuum. Within the purview of this research, an elucidation is offered concerning the particular efficacy of new fluorophore TPEF as a superior staining modality for PS nanoplastics. Consequently, the salient dye affinity characteristics of **TPEF is suitable** for the methodologies aimed at the tracing of fluorescent nanoplastic entities. 

## Figures and Tables

**Figure 1 molecules-28-07102-f001:**
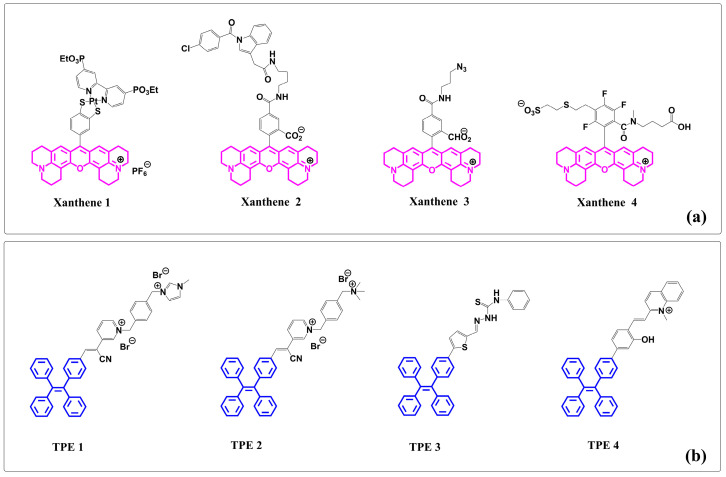
(**a**) Representative structures of xanthene- or TPE-based fluorophore compounds from recent publications: (i) xanthene 1, (ii) xanthene 2, (iii) xanthene 3, and (iv) xanthene 4. (**b**) TPE–containing fluorophore compounds: (i) TPE 1, (ii) TPE 2, (iii) TPE 3, and (iv) TPE 4. (Note: Structures of xanthene are shown in pink and TPE structures are shown in blue.)

**Figure 2 molecules-28-07102-f002:**
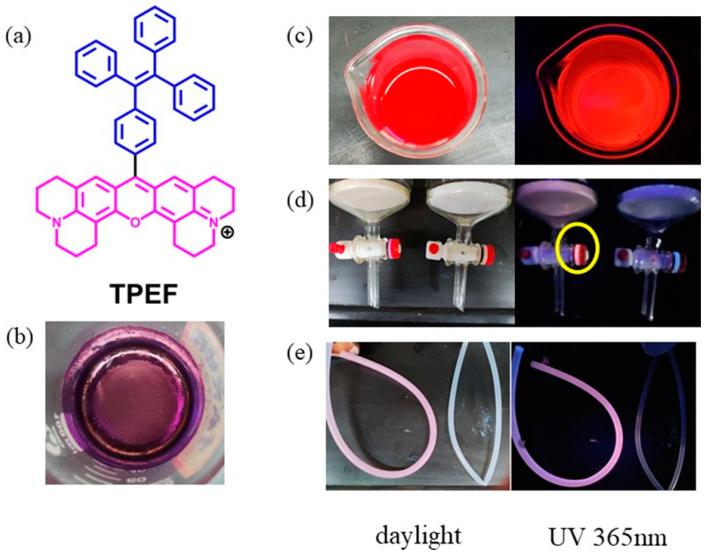
(**a**) Molecular structure of TPEF (Note: Structures of xanthene are shown in pink and TPE structures are shown in blue.); (**b**) sample of TPEF as a purple solid; (**c**) TPEF methanol solution; (**d**) plastic gasket on the glass column (left: after exposure to TPEF methanol solution (Note: The yellow circle is used to highlight the area of the plastic gasket that has been stained by TPEF, showing increased fluorescence.); right: without exposure to TPEF methanol solution); and (**e**) plastic tube (left: after exposure to TPEF methanol solution; right: without exposure to TPEF methanol solution).

**Figure 3 molecules-28-07102-f003:**
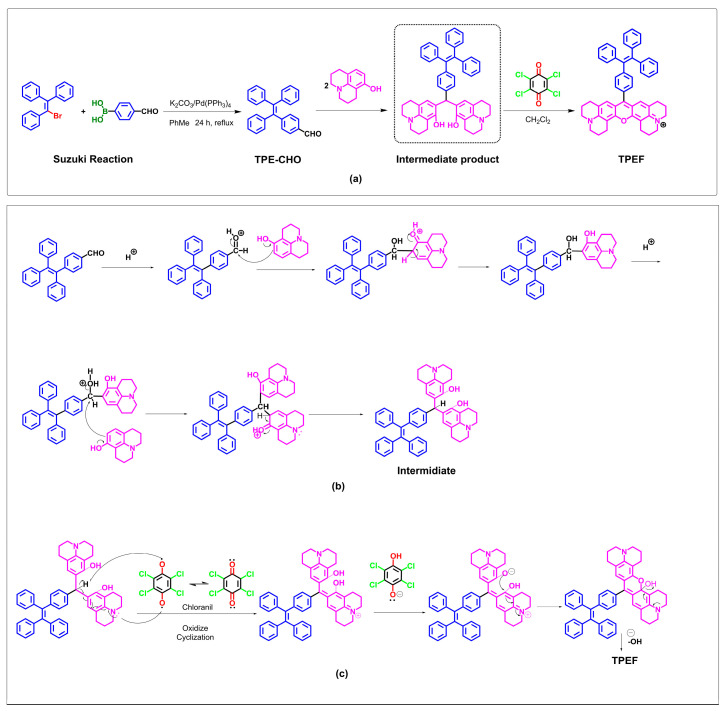
Synthesis and the mechanistic hypothesis for TPEF formation from TPE–CHO. (**a**) Synthesis of TPE–CHO and TPEF; (**b**) complete mechanism of TPEF formation from TPE–CHO; and (**c**) mechanism of oxidize cyclization.

**Figure 4 molecules-28-07102-f004:**
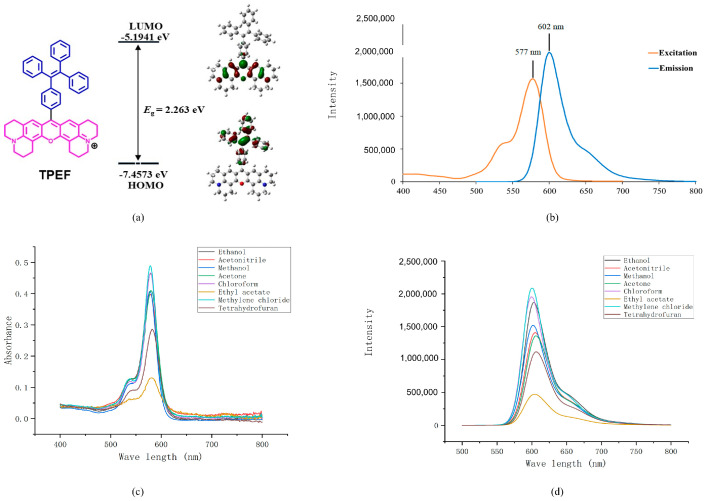
(**a**) Molecular orbital amplitude plots showing the highest occupied molecular orbital (HOMO) and lowest unoccupied molecular orbital (LUMO) energy levels, along with the HOMO/LUMO energy gap of TPEF. (**b**) Excitation/emission fluorescence spectra of TPEF in MeOH (0.1 mmol L^−1^). (**c**) UV–vis absorption of TPEF in various solvents (0.1 mmol L^−1^). (**d**) Fluorescence spectra of TPEF in different solvents (0.1 mmol L^−1^). Details of the UV absorption peak (nm), emission peak (nm), and Stokes shift (nm) for TPEF in all the mentioned solvents, including ethanol, MeCN, methanol, acetone, CHCl_3_, ethyl estate, CH_2_Cl_2_, and THF, are provided in Appendix A.

**Figure 5 molecules-28-07102-f005:**
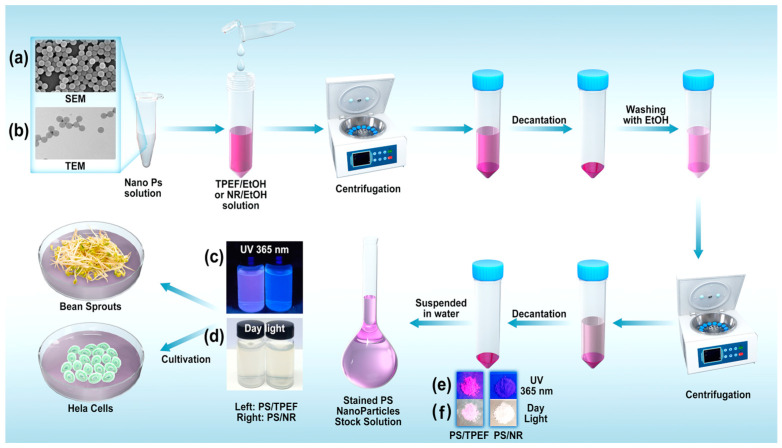
Representative SEM (**a**) and TEM (**b**) images of PS particles and the staining procedure used in this study (note: additional images detailing the particles before and after the TPEF staining can be found in the Appendix A). (**c**) Stained PS particles suspended in water under 365 nm UV irradiation (left: TPEF staining; right: NR staining). (**d**) Stained PS particles (left: TPEF-stained; right: NR-stained) suspended in water in daylight. (**e**) Dry stained PS particles (left: TPEF-stained; right: NR-stained) under 365 nm UV irradiation. (**f**) Dry stained PS particles (left: TPEF-stained; right: NR-stained) in daylight.

**Figure 6 molecules-28-07102-f006:**
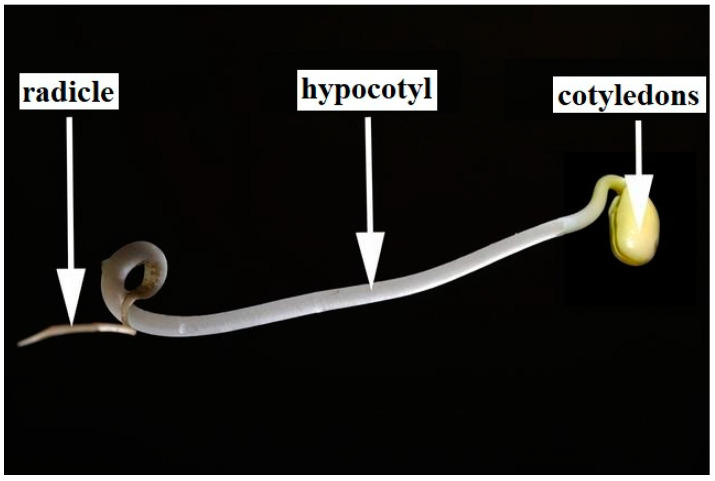
Anatomy of a soybean sprout.

**Figure 7 molecules-28-07102-f007:**
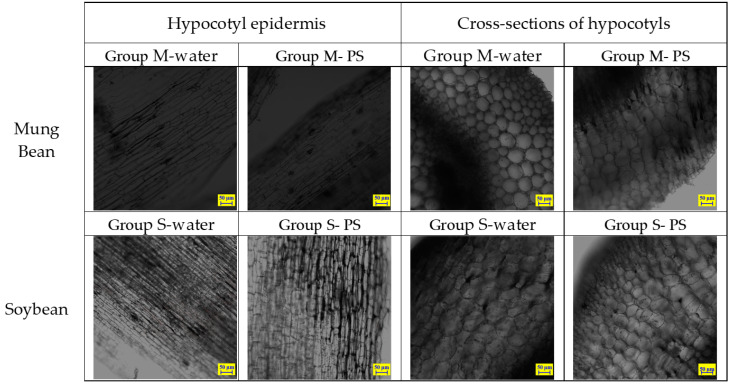
CLSM images of control mung bean sprouts and soybean sprouts samples on day 5.

**Figure 8 molecules-28-07102-f008:**
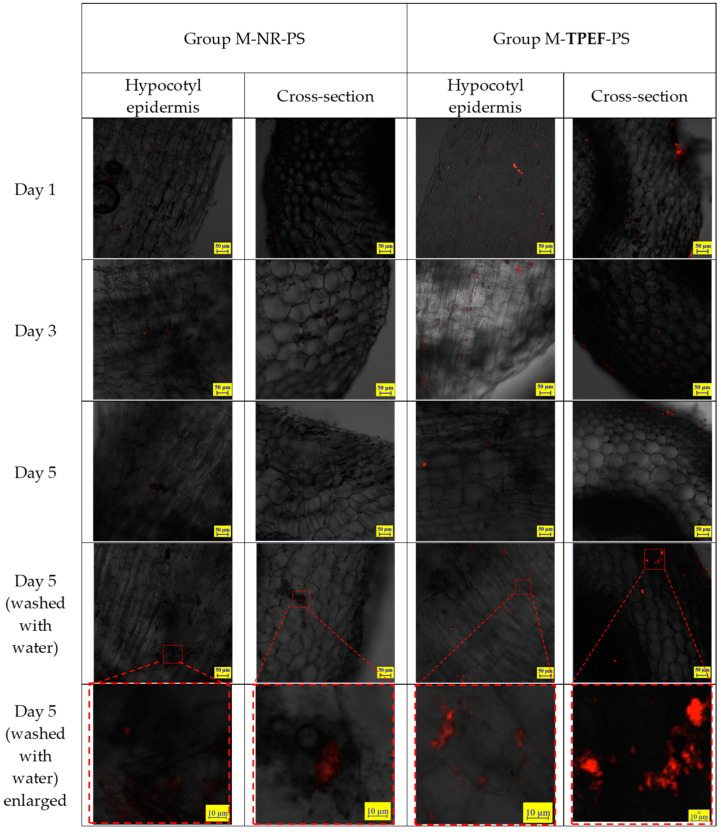
CLSM images of hypocotyl epidermis in test samples of mung bean sprouts.

**Figure 9 molecules-28-07102-f009:**
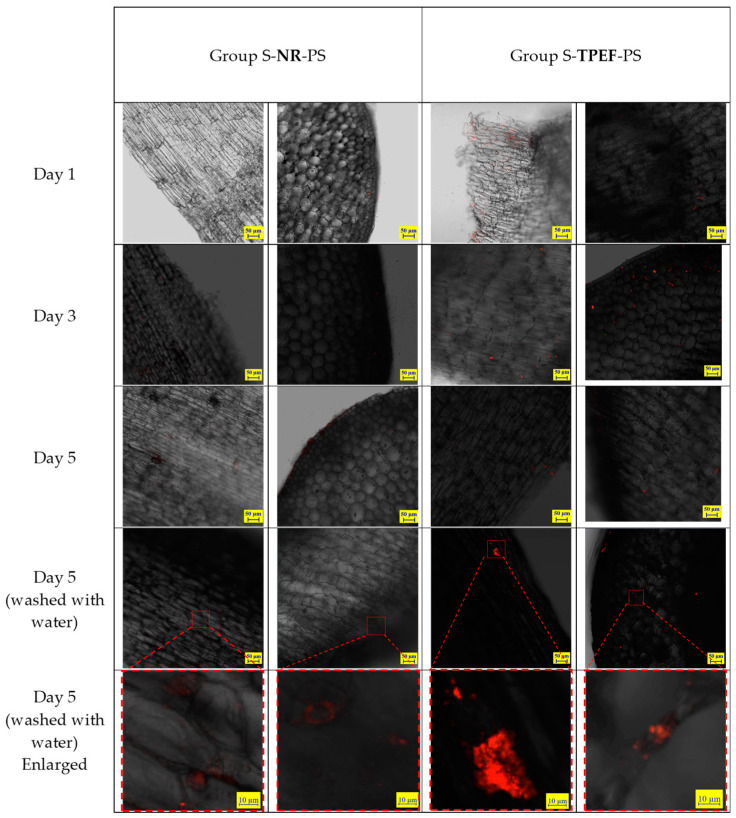
CLSM images of hypocotyl epidermis in test samples of soybean sprouts.

**Figure 10 molecules-28-07102-f010:**
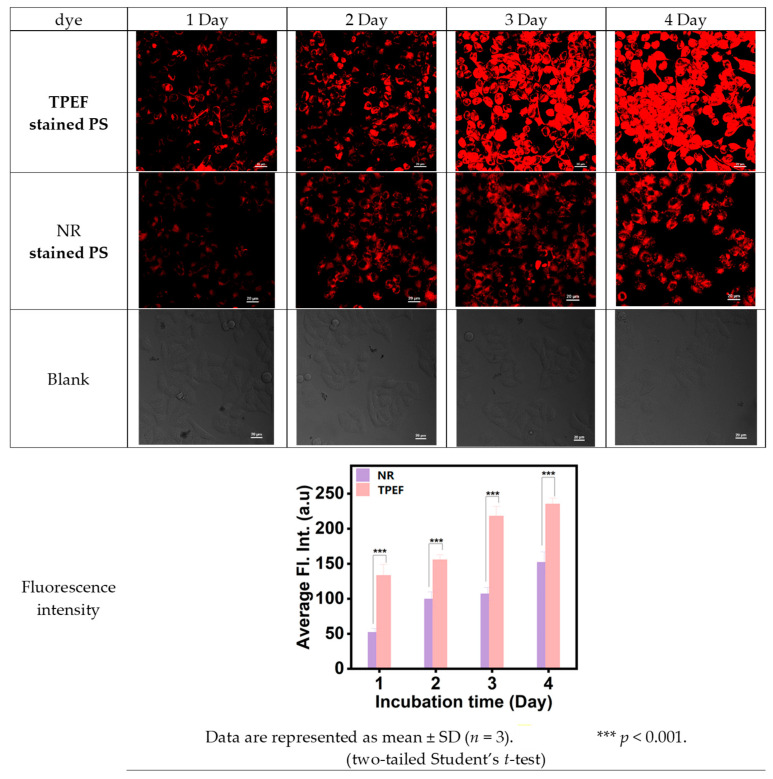
CLSM images of HeLa cell test samples and the blank samples for comparison.

**Table 1 molecules-28-07102-t001:** Mung bean sprouts and soybean sprouts utilized in this project.

Bean Sprouts	Groups	PS Size (nm)	PS (µg·mL^−1^)	Dye
Mung bean sprouts	Group M–water	None	0	–
	Group M– PS	80	100	–
	Group M–TPEF–PS	80	100	TPEF
	Group M–NR–PS	80	100	NR
Soybean sprouts	Group S–water	None	0	–
	Group S–PS	80	100	–
	Group S–TPEF–PS	80	100	TPEF
	Group S–NR–PS	80	100	NR

## Data Availability

All the data are in the manuscript and Appendix A.

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
