# Peer review of "New Fluorophore and Its Applications in Visualizing Polystyrene Nanoplastics in Bean Sprouts and HeLa Cells"

_molecules, 2023, doi:10.3390/molecules28207102_

Round 1
Reviewer 1 Report
In this submitted manuscript, Xing and coworkers are reporting an interesting TPEF-Xanthene hybrid fluorophore for visualizing nano plastics in both human and plant cells via fluorescence microscopy. The scope of the manuscript seems to be interesting to the bioimaging community. In overall, this will be publishable eventually with following revision.
(1). In the introduction, authors should highlight the importance of the TPEF-based fluorophore systems in imaging applications. Also, I am suggesting authors to prepare a scheme in the introduction while including structures with recently reported TPEF-based hybrid dye systems for biological application. Must include the appropriate references.
(2). Briefly highlight the importance of the bioimaging applications with small-molecule probes, especially with red to near-infrared emitting probes. Some of the recommended references to include: Chemical Communications 58.71 (2022): 9855-9869; Topics in Current Chemistry 380.4 (2022): 22; Molecules 27.23 (2022): 8421; Photochemical & Photobiological Sciences (2022): 1-15.
(3). Authors should summarize the photophysical properties of the probe in different solvents including absorbance, emission, Stokes’ shift, fluorescent quantum yield, and molar absorptivity in a table form and must include in the manuscript.
(4). The response of the probe for pH must be studies.
(5). Authors should conduct photophysical studies to demonstrate the aggregation possibility of the probe while using an appropriate solvent mixture system.
(6). The provided abstract must be revised carefully. The current version does not highlight the significance of this work.
(7). Based on the computational HOMO-LUMO calculations, authors should calculate the excitation wavelength for the probes and then must compare with the experimental values.
(8). Please provide the synthetic yields in the reaction schemes. The characterization data must be provided for all isolated intermediate compounds.
(9). Please provide the ppm error for the HRMS data.
(10). I think it is appropriate to convert table 2 into a figure instead.
(11). For all imaging studies, authors should clearly mention excitation wavelength and emission collection method while stating the probe concentration clearly in the figure captions.
(12). Authors should conduct colocalization experiments for mitochondria for the HeLa cell studies.
(13). There are several typo ad grammatical errors in the manuscript. Please revise carefully.
(14). Revise the conclusion carefully as there are textual and grammatical errors.
There are several typos, grammatical and textual errors. Authors should pay close attention during the revision.
Reviewer 2 Report
This study explores the use of a novel fluorophore, Tetraphenylethylene Functional Group with a Fused Xanthene (TPEF) and its application for nanoplastic detection in bean sprouts and HeLa cells. The results and discussion begin with TPEF synthesis, followed by its spectral properties and DFT calculations. Then, the resulting dye was used to stain PS nanoplastics with a diameter of 80 nm. This was done to study nanoplastic impacts on the growth of bean sprouts and HeLa cells. Also, the authors compared TPEF's performance with Nile Red as a reference dye and demonstrated the advantages and novelty of TPEF for staining nanoplastics in biological samples. This investigation concludes that nanoplastics can penetrate both bean sprouts and human HeLa cells. I have some major concerns that I address in the following paragraphs. The language should be carefully revised. There are several typos.
Major remarks:
There are many very general parts in the introduction, including plastic production and landfill formation etc. The introduction needs to be shortened or a few general parts should be eliminated. Figure 1's caption also fails to explain the figure properly. There is also no clear explanation of why some images (Figure 1b) are shown without scale. C is highlighted in the caption, but not in the figure. Additionally, Figure 1a is taken directly from Gillibert et al., ES&T,2019 without being cited. For me, even without Figure 1, the message is clear.
The second major concern I have is the title of the manuscript. This gives the impression that the proposed dye could be useful for all nanoplastics. However, the authors only show that PS can be labelled. The authors should prove that TPEF can also be used to label other polymers, such as PE or PP, which are the biggest contributors to landfills.
In addition, confocal images of PS nanoplastics in beans and HeLa cells do not show uniform brightness or contrast, making comparison difficult (Table 2 and 3). Also, there is much less fluorescence intensity in beans sprouts than in HeLa cells. It is not clear why nanoplastics uptake in Hela cells is higher than in beans. It is also necessary to discuss how time affects of NP uptake in bean sprout cells as well as Hela cells in connection with the mechanisms of Nanoplastics uptake.
Minor remarks:
Line Number 49-52, Line Number 54-56, line number 58-60 and line lumber 61-62 : Citation is required.
Line Number 89-92: "Nile Red, for example, fails to detect microplastics with dimensions below 3 μm or above 5 mm, and exhibits inadequate specificity, inadvertently staining non-microplastic organic matter." Please justify this size dependent labelling along with proper citations. I disagree with this statement because there is already some literature available regarding the labeling of nanoplastics with Nile Red and RhB.
Line number 130: Please change to Figure 2 a
Figure 5 - Scales for the SEM and TEM images are missing.
Line Number 275: Please indicate the concentration of the NR or TPEF stock solution in ethanol. Please also mention the concentration of PS latex particles before mixing with dye solution. How do you verify that there is no excess dye left after multiple washings ?
Line Number 276 : Instead of "multiple washing" please give a number
Line number 277 : Is there any reason to set PS particles concentration as 100 µg/ml ?
Line number 316 : why does NP accumulate in the middle of the hypocotyl?
Table 2 : Anatomy of a soybean sprout should be separated from the confocal images by a separate figure or table.
Table 2 represents the confocal images of the control sample containing no fluorescence dye at all. In that case, how do you create images of bean cells without fluorescence? I assume CLSM is not fluorescence mode. Please comment on that in the materials and methods section.
Line number 331 : Please change "section" to "cross-sections of the hypocotyls"
Table 3 : What is the reason for the darker images than those in Table 2? Images of the hypocotyl epidermis (Group M-NR-PS and Group M-TPEF-PS) are significantly different from those of the control sample (Group M-PS). I expected the same image as Group M-PS in table 3, with some bright spots due to NR and TPEF fluorescence. Do you get the same images from replicate to replicate? Please describe the distribution of nanoplastics within the hypocotyl and how the structure of beans appear at various locations within the hypocotyl.
Line number 345-349 :"The CLSM images also demonstrated that the TPEF-stained PS nanoparticles had a more uniform distribution and smaller size than the NR-stained PS nanoparticles. This discrepancy could be attributed to the different dyeing mechanisms and aggregation behaviors of TPEF and NR on the PS particle surface". In both cases one cannot expect a difference in the dyeing mechanism as it is driven by hydrophobic interactions. There are also only a few particles in each image, so perhaps the discrepancy is simply the result of poor statistics in this case.
Line number 371-381 : This paragraph is repeated in section 2.5 as "Detection of Nanoplastics in Soybean Sprouts". I would suggest the authors delete section 2.5 and move the paragraph (371-381) above line number 364 which makes reading much easier.
Line Number 382 : "Possible Mechanisms of Nanoplastics Uptake by Bean Sprouts" : In this section, authors should explain why only a few nanoplastics are uptaked in both bean sprouts compared to HeLa cells. After 5 days, nanoparticle uptake does not significantly change for bean sprouts, indicating that diffusion by osmosis is not the driving force. In this case, one might observe an increase in uptake over time.
Line Number 432 : "Table 5: CLSM images of HeLa cell test samples". Please add a control sample with non-stained PS nanoplastics in HeLa cells at different times as shown in Table 5. Please also explain why the nucleus size is smaller when the NP is labeled with NR in Table 5.
Line number 428 and 430 : Please change Table 4 to Table 5
Line Number 504-505:"Sonicated for 30 minutes at 60 degrees Celsius". Why is this step necessary? Additionally, I would request that the authors include images of stained or non-stained PS latex particles after the entire staining process is complete, as described in 3.4.1. In order to ensure that the PS NP used for the investigation does not change in size or shape as a result of the staining process, a control sample is needed.
Line Number 522-526 : Confocal Imaging of Nanoplastics : Please add the type (water or oil immersion) and magnification of the objective used. Also include the wavelength of the incident laser beam used for imaging.
Line Number 533-534 : "Fluorescence images were promptly captured by Nikon single particle microscopy with a 40× objective lens". Table 5 is taken with CLSM or single particle microscopy?
Round 2
Reviewer 1 Report
Authors have have performed all my recommended revisions in the re-submitted manuscript. Therefore, I recommend to accept it in present form with a final English language check.
Minor English text/grammar check is highly recommended.
Reviewer 2 Report
Comments to the authors
The authors have considered and responded appropriately to my comments. There are still typos and incorrectly numbered tables and figures. I would advise the authors to review the manuscript carefully to verify the table numbers and figures. After minor revisions, the manuscript can be accepted.
This revised graphic abstract is unclear. I found the previous graphic abstract clearer. Please use that version if possible.
I think it would be better if paragraphs 1 and 2 were combined.
In line 39, insert the references before the full stop: "as carriers for harmful agents".
Add references [6, 7] and [8, 12- and 13-15] before the full stop.
Line 50, add a full stop between reference 17 and Fluorescent probes "between biological samples [17] Fluorescent probes
Authors use tables and figures at different times. In order to maintain consistency, it is best to replace all tables except table one with figures or vice versa. In the manuscript after Table 1, the following table number 3 and 2 is missing, whereas in the test Table 2 is cited
One of the confocal images in Figure 7 has a problem (Group M-PS, Soybean).
Table 5's caption should include a description of the blank sample.
As the control images in Table 5 do not contain a fluorescent probe, it is better to display confocal images in reflection mode to view some domains of hela cells (similar to Table 7's confocal images).
Figure S1, Figure S2 and Figure S5 are not mentioned in the manuscript but cited in the supplementary information
